# A Candidate Ac_3_-S-LPS Vaccine Against *S. flexneri* 1b, 2a, 3a, 6, and Y Activates Long-Lived Systemic and Mucosal Immune Responses in Healthy Volunteers: Results of an Open-Label, Randomized 2 Clinical Trial

**DOI:** 10.3390/vaccines13030209

**Published:** 2025-02-20

**Authors:** Vladimir A. Ledov, Victor V. Romanenko, Marina E. Golovina, Biana I. Alkhazova, Alexander L. Kovalchuk, Petr G. Aparin

**Affiliations:** 1ATVD-Team Co, Ltd., 115522 Moscow, Russia; ledov_va@sysbiomed.ru (V.A.L.); romanenko.v47@gmail.com (V.V.R.); golovina@atvd-team.ru (M.E.G.); alkhazova@atvd-team.ru (B.I.A.); kovalchuk_a@atvd-team.ru (A.L.K.); 2Research Institute for Systems Biology and Medicine, 117246 Moscow, Russia; 3Laboratory of Carbohydrate Vaccines, National Research Center-Institute of Immunology, Federal Medical Biological Agency of Russia, 24, Kashirskoe Shosse, 115478 Moscow, Russia

**Keywords:** *Shigella flexneri*, vaccine, lipopolysaccharide

## Abstract

Objectives: Determination of reactogenicity and immunogenicity of a pentavalent candidate vaccine against *S. flexneri* 1b, 2a, 3a, 6, and Y (PLVF). Methods: The study involved 80 healthy adult volunteers aged 18–55 years. Groups were subcutaneously immunized twice at a 30-day interval with 62.5 μg/0.5 mL or 125 μg/0.5 mL of the vaccine. Results: During the entire 8-month period of post-vaccination observation, the vaccine was well tolerated, with no local or systemic reactions detected objectively. The results of laboratory studies demonstrated no effect on the main indicators of hemogram, biochemical blood test, or urinalysis. IgA, IgG, and IgM levels against LPS *S. flexneri* 1b, 2a, 3a, 6, and Y were examined before vaccination, a month after each vaccination, and 6 months after booster vaccination. One month after vaccination, IgA and IgG seroconversions were observed in 67.5–82.5% (depending on serotype) and 60–77.5% of volunteers, respectively. Booster immunization did not have a significant effect on vaccine immunogenicity. In two separate groups of 15 and 9 volunteers for mucosal sIgA, IgA, and IgG titer determination after immunization with a 125 μg vaccine dose, paired stool, and saliva samples were taken before and one month after vaccination. In 26.7–40% of volunteers, there was a 2-fold and higher increase in sIgA titer for the studied serotypes in the feces and in 66.7–88.9% in saliva. IgA and IgG 2-fold conversion rates were 26.7–53.3% and 33.3–46.7% in the feces, 33.3–77.9%, and 66.7–77.8% in saliva, respectively. Conclusions: the tolerability of PLVF and the pronounced humoral immune response allow us to proceed to the phase 3 clinical trial stage.

## 1. Introduction

Shigellosis is a worldwide problem. *Shigella* infection is the second leading cause of diarrheal mortality in children under 5 years of age (12.9%) and the leading cause of death in adults over 70 years of age (10.7%) worldwide [1]. Although there is no readily available and rapid diagnostic test for shigellosis, WHO recommends treating suspected *Shigella*-associated episodes with antibiotics [2]. Antibiotic use reduces the incidence of infection; however, their overuse increases antimicrobial resistance [2,3,4,5,6,7]. The global rise in antibiotic resistance makes the development of vaccines against *Shigella* an important global health priority [8].

The current classification divides the genus *Shigella* into four species based on serotyping: *S. dysenteriae*, *S. boydii*, *S. flexneri*, and *S. sonnei* [9]. These species are further subdivided into serotypes and subserotypes based on the specificity of the repeating polysaccharide units that form part of the O-antigen (OAg) of lipopolysaccharide (LPS). Currently, *S. dysenteriae* species is known to include 14 serotypes, *S. boydii* has 19 serotypes, *S. flexneri* has 15 serotypes and subserotypes, and *S. sonnei* has only one serotype [10]. *Shigella* serotypes are distributed heterogeneously across different regions and countries. This implies that to achieve global coverage, a polyvalent and cross-protective *Shigella* vaccine is needed.

*S. flexneri* causes a typical mucosal infection. *Shigella* invades the epithelium of the colon and rectum of primates and humans, causing acute inflammation of the mucosa. Damage to the colonic mucosa leads to clinical symptoms such as diarrhea, abdominal pain, and cramps. Severe intestinal damage by infection leads to abscesses and ulcers. In the absence of effective treatment, patients with shigellosis may develop secondary complications such as septicemia, pneumonia, and hemolytic uremic syndrome [11].

According to GEMS (Global Enteric Multicenter Study), a vaccine consisting of *S. sonnei* and *S. flexneri* 2a, 3a, and 6 OAgs can provide protection against shigellosis by 64% globally. This may increase to ~85% due to cross-protection against heterologous serotypes of *S. flexneri* [12]. We used a method for partial removal of fatty acids from the lipid A of S. *flexneri* LPS to produce a non-toxic immunogen: triacyl long-chain modified LPS (Ac_3_-S-LPS) from epidemically significant serotypes *S. flexneri* 1b, 2a, 3a, 6, and Y [13].

The present phase 2 clinical study is devoted to all aspects of the safety and reactogenicity of the pentavalent LPS candidate vaccine against *S. flexneri* 1b, 2a, 3a, 6, and Y (PLVF). It was conducted in healthy adult volunteers using two-time subcutaneous immunization with different doses of the vaccine preparation (62.5 µg and 125 µg) with an interval of 30 days. We also conducted a study of the systemic and mucosal immune responses to each of the vaccine’s Ac_3_-S-LPS antigens.

## 2. Materials and Methods

### 2.1. Vaccine Preparation

The candidate vaccine against *S. flexneri* 1b, 2a, 3a, 6, and Y was produced by ATVD-Team Co., Ltd. (Moscow, Russia) and formulated as previously described [14]. Each ampoule contained 0.125 μg *S. flexneri* 1b, 2a, 3a, 6, and Y Ac_3_-S-LPS (0.025 μg each) dissolved in 0.5 mL sterile isotonic buffer.

### 2.2. Study Design

This was an open, randomized multicenter clinical trial conducted in the Russian Federation from April to December 2017. Volunteers were recruited in the municipal budgetary institution “Central city hospital No. 7”, 620137, Yekaterinburg, and in the St. Petersburg state health care institution “City hospital No. 40 of the Kurortny administrative district” 197706, Sestroretsk.

At the screening stage, 86 volunteers aged 18–55 years signed the informed consent. All volunteers who passed the screening and were included in the study were randomized using the sealed envelope system into two even groups: one group received 62.5 µg of the vaccine preparation and the other 125 µg. The vaccine was administered subcutaneously into the upper third of the shoulder twice with an interval of 30 days. On the day of vaccination, volunteers were admitted to the inpatient department of the clinical center for no longer than 24 h. The study procedures are presented in Table 1. Only those who met all the inclusion criteria and none of the exclusion criteria were eligible for enrollment. Key exclusion criteria were participation in any other clinical trial; pregnancy and breastfeeding; exacerbation of allergic diseases; acute or exacerbation of chronic infectious diseases (including HIV, syphilis, hepatitis B, and C); acute somatic diseases; or exacerbation of chronic somatic diseases.

### 2.3. Safety Evaluation

Participants were monitored for safety throughout the study (from the moment vaccine preparation was administered) using the following parameters: frequency and type of AEs and changes in laboratory parameters. General reactions that were assessed included increased body temperature, high blood pressure, tachycardia, malaise, headache, nausea, vomiting, diarrhea (noting the frequency and consistency of stool), abdominal pain, and other symptoms, as well as deviations from normal levels in the results of the complete blood count and biochemical blood test. Local reactions that were assessed included pain, redness, and swelling at the injection site, the occurrence of infiltrates, pain, and enlargement of regional lymph nodes.

### 2.4. Collection of Samples and Immunogenicity Evaluation

#### 2.4.1. Blood Specimens

Venous blood was collected during screening, one month after vaccination, one month after revaccination, and 6 months after revaccination. Serum was separated immediately after blood collection and frozen at −70 °C until assayed by ELISA. Serum levels of IgA, IgG, and IgM to *S. flexneri* 1b, 2a, 3a, 6, and Y LPS were determined using direct ELISA (standard protocol) with native *S. flexneri* LPS (phenol–water extraction, protein and nucleic acid content < 0.5%) *S. flexneri* 1b, 2a, 3a, 6, Y adsorbed on microplates (Greiner, Kremsmünster, Austria) [14] and secondary mouse anti-human IgG, IgM, or IgA conjugated with HRP (Sigma-Aldrich, St. Louis, MO, USA).

#### 2.4.2. Fecal and Saliva Samples

Paired stool samples from 15 volunteers and saliva samples from 9 volunteers vaccinated with 125 μg of the preparation were collected before vaccination and one month after vaccination. Saliva samples were collected by passive drooling in 8.0 mL sterile penicillin vials in the morning before meals after rinsing the mouth with a physiological solution. After centrifugation (1000× *g* for 5 min), the supernatant was placed in polystyrene tubes (1.5 mL) and immediately frozen at −70 °C.

Approximately 1.0 g of feces were placed in sterile glass vials with 5.0 mL of sterile saline. The samples were left to stand on the bench at room temperature for 15 min with intermittent shaking. After thorough mixing, the extracts were passed through a filter paper. The filtrate was placed in polystyrene tubes (1.5 mL) and immediately frozen at −70 °C.

On the day of assay, samples were thawed and centrifuged (10,000× *g* for 5 min) prior to experiments. *Shigella* LPS-specific IgA, sIgA, and IgG were determined in coprofiltrates and in the supernatant fraction of saliva using direct ELISA (standard protocol). For detection of sIgA, we used mouse anti-human sIgA and then goat anti-mouse IgG conjugated to HRP (Sigma-Aldrich, St. Louis, MO, USA).

### 2.5. Randomization and Blinding

This study was open, but with blinding of the volunteer randomization data for the personnel of the laboratory where the immunological studies were conducted.

Volunteer randomization was performed only after the investigator assessed all inclusion/exclusion criteria, including laboratory tests and confirmation of the volunteer’s compliance with all inclusion criteria of the study protocol.

Randomization was carried out in groups for vaccination at doses of 62.5 µg or 125 µg (0.5 mL) in a ratio of 1:1 regardless of age and gender. Each volunteer selected for participation in the study was assigned an individual code upon completion of screening. It consisted of a two-digit number of the research center and a four-digit serial number in the study; for example, a volunteer in research center 1 with serial number 25 was assigned code 010025. Upon completion of the study (after the last visit of the last included volunteer), a representative of the sponsor performed a final check of the correctness in filling out the information registration card by the investigators. If the representative of the sponsor had no questions for the investigators, statistical processing of the data began. The randomization code was opened at the end of the study.

### 2.6. Analyses

Demographic and anthropometric characteristics and clinical analysis results are presented as mean ± SD. The immune response in volunteers is represented by a ≥4-fold increase in specific antibodies titers in serum (seroconversion) and a ≥2-fold and ≥4-fold increase in specific antibodies titers in saliva and coprofiltrates, compared to baseline. The titer was determined using cut-off = M (blank, BSA adsorbed on plate) + 3SD.

## 3. Results

### 3.1. Prestudy Screening

Based on a physician’s examination and laboratory tests, 80 volunteers aged 18–55 years were included in the study (mean age 44.5 ± 9.4 years). The first informed consent form was signed on 3 April 2017. The date of completion of the study by the last participant was 29 December 2017. There were no withdrawals of study participants after randomization. The study was completed due to the full implementation of the clinical trial plan. Group 1: 40 volunteers who were injected with the vaccine twice subcutaneously in the upper arm with an interval of 30 days at a dose of 62.5 µg; Group 2: 40 volunteers who were vaccinated with the vaccine twice subcutaneously in the upper arm with an interval of 30 days at a dose of 125 µg. There was no difference in baseline demographic data between the two groups (Table 2).

### 3.2. Safety and Clinical Adverse Events

During the entire period of post-vaccination observation, no local or systemic reactions were observed objectively (based on examination by the physician–investigator) or subjectively (based on the self-observation diaries). There were no reports of serious adverse events among the study participants. The results of the laboratory studies show that the subcutaneous administration of the vaccine twice with an interval of 30 days to healthy volunteers aged 18–55 years does not affect hematological and biochemical parameters or urine composition. The observed minor deviations from normal values of some blood and urine parameters did not have clinical manifestation and were explained by physiological reasons (Appendix A).

### 3.3. Human Systemic Humoral Shigella LPS-Specific Immune Response After PLVF Immunization

The primary immune response one month after the first immunization with PLVF at doses of 62.5 µg and 125 µg was characterized by the occurrence of 4-fold seroconversions of IgG, IgA, and IgM to all five components of the vaccine. A slightly higher level of seroconversions was recorded in the response against the LPS serotype *S. flexneri* 2a. Revaccination one month later with the corresponding doses of the vaccine did not lead to the induction of a secondary immune response and an increase in seroconversion levels.

However, the most interesting data were obtained by us on the long-lived systemic immune response. Six months after booster immunization with PLVF, anti-*S. flexneri* 1b IgG seroconversion was 40.0% for group 1 and 60.0% for group 2, anti-*S. flexneri* 2a—70.0% for group 1 and 70.0% for group 2, anti-*S. flexneri* 3a—52.5% for group 1 and 67.5% for group 2, anti-*S. flexneri* 6—60.0% for group 1 and 57.5% for group 2, anti-*S. flexneri* Y—60.0% for group 1 and 57.5% for group 2. Anti-*S. flexneri* 1b IgA seroconversion was 35.0% for group 1 and 60.0% for group 2, anti-*S. flexneri* 2a—67.5% for group 1 and 72.5% for group 2, anti-*S. flexneri* 3a—52.5% for group 1 and 67.5% for group 2, anti-*S. flexneri* 6—50.0% for group 1 and 60.0% for group 2, anti-*S. flexneri* Y—42.5% for group 1 and 67.5% for group 2. In all cases, the percentage of seroconversion in the second group was generally higher than in the first. No significant difference was found between seroconversion after the first and repeated vaccinations (Table 3).

### 3.4. Human Polyvalent Mucosal Shigella LPS-Specific Salivary and Fecal sIgA, IgA, and IgG Response After PLVF Immunization

One month after a single immunization with 125 µg PLVF 26.7%, 40%, 40%, 40%, and 53.3% of participants showed ≥2-fold increase and 26.7%, 33.3%, 33.3%, 26.7%, and 46.7% of participants showed ≥4-fold increase in titers of fecal IgA specific to *S. flexneri* 1b, 2a, 3a, 6, and Y LPS, respectively.

A ≥2-fold titer of fecal sIgA specific to *S. flexneri* 1b, 2a, 3a, 6, and Y was detected in 26.7%, 33.3%, 40%, 33.3%, and 40% of participants, respectively; ≥4 increases in 26.7%, 33.3%, 33.3%, 13.3%, and 26.7% of participants, respectively. A ≥2-fold rise in titers of fecal IgG specific to *S. flexneri* 1b, 2a, 3a, 6, and Y was detected in 33.3%, 40%, 46.7%, 33.3%, and 46.7% of participants, respectively; and a ≥4-fold rise in 20%, 33.3%, 26.7%, 26.7%, and 40% of participants, respectively (Table 4). This finding suggests transport and transudation of IgG into the intestinal lumen from the systemic circulation.

In saliva of vaccinated volunteers after primary vaccination, a ≥2-fold increase in titers of IgA specific to *S. flexneri* 1b, 2a, 3a, 6, and Y was detected in 44.4%, 55.5%, 33.3%, 33.3%, and 77.8% of participants, respectively; a ≥4-fold increase was in 11.1%, 33.3%, 22.2%, 33.3%, and 33.3% of participants, respectively.

The frequency of increases of sIgA and IgG titers in saliva was higher than in feces. In saliva of vaccinated volunteers after primary vaccination, a ≥2-fold increase in titers of sIgA specific to *S. flexneri* 1b, 2a, 3a, 6, and Y was detected in 66.7%, 77.8%, 77.8%, 88.9%, and 88.9% of volunteers, respectively; a ≥4-fold increase in 22.2%, 44.4%, 44.4%, 44.4%, and 55.5% of volunteers, respectively. An increase of ≥2-fold increase of titers of IgG specific to *S. flexneri* 1b, 2a, 3a, 6, and Y was detected in 77.8%, 77.8%, 66.7%, 77.8%, and 66.7% of volunteers, respectively; ≥4-fold increase in 22.2%, 55.5%, 44.4%, 44.4%, and 44.4% of volunteers, respectively (Table 4).

## 4. Discussion

Based on data obtained from controlled challenge trials in volunteers, epidemiological studies, and studies of primates, an exceptional role of LPS in the development of a protective immune response against *Shigella* has been established [15,16,17,18].

Ac_3_-S-LPS, the active substance of the pentavalent vaccine PLVF, demonstrated a high safety profile after subcutaneous administration with doses of 62.5 µg and 125 µg in a phase 2 clinical study. During the entire period of clinical observation, no local and/or systemic adverse reactions were detected in any study participant objectively (based on examination by the investigator) or subjectively (based on patient complaints). Changes in the results of the complete blood count, biochemical blood test, and urinalysis performed during the study were assessed as clinically insignificant and independent of vaccination, which indicates the safety of the PLVF vaccine.

The reactogenicities of guaBA-based live attenuated candidates from the *S. flexneri* 2a 2457T strain [19], virG and iuc deletions from the *S. flexneri* 2a 494 wild type strain [20], and the Ty21a typhoid vaccine displaying *Shigella* LPS [21] are rather high despite the oral route of administration.

Subunit or outer membrane vesicle (OMV) LPS-enriched vaccines InvaplexNAT (*S. flexneri* 2a) [22], InvaplexAR-Detox (*S. flexneri* 2a) [23], OMV (GMMA) (*S. flexneri* 1b, 2a, 3a, and *S. sonnei*) [24], which do not contain LPS with hexa-acyl lipid A, are significantly safer. However, they still contain LPS with penta- and tetra-acyl lipid A and, therefore, can cause temperature increases up to 38 °C, headache, pain, neutropenia, and local reactions [25].

Apparently, the technology of controlled chemical detoxification of native LPS of *S. flexneri* used to produce a safe vaccine candidate PLVF based on Ac_3_-S-LPS allows us to obtain drugs comparable in safety to OAg-containing vaccines that do not have the lipid A domain [14]. However, the high level of safety of OAg of *S. flexneri* (O-polysaccharide) is compromised by the need to introduce into the vaccine carrier proteins that can be attached chemically (*S. flexneri* 2a synthetic OAg conjugate with tetanus toxoid [26], natural *S. flexneri* 2a and *S. sonnei* OAg conjugate with tetanus toxoid [27]) or using bioconjugation technology (*S. flexneri* 2a, 3a, 6, and *S. sonnei* OAg bioconjugates with *Pseudomonas aeruginosa* recombinant exoprotein A [28]). In the last case, increased protein load may influence the allergenic profile characteristics of the preparation.

A characteristic feature of the immunogenicity of our pentavalent Ac_3_-S-LPS vaccine, in contrast to other polyvalent candidate vaccines against shigellosis flexneri, is the induction of IgA in volunteers to each of the vaccine antigens. Reports on trials of tetravalent *S. flexneri* 1b, 2a, 3a, and *S. sonnei* OMV (GMMA) [24], *S. flexneri* 2a, 3a, 6, and *S. sonnei* bioconjugate vaccine [28] do not contain information on the induction of an IgA response in vaccinated individuals.

After repeated immunization groups of volunteers with the same dose, no booster effect was observed, which is comparable with the data obtained with immunization with a semisynthetic conjugate *S. flexneri* 2a vaccine [26], and four-component GMMA-based vaccines [24]. The assumption that the powerful primary immune response of volunteers to different types of dysentery vaccines is due to preexisting immune memory for *Shigella* antigens is becoming increasingly substantiated.

At the same time, we recorded the activation of a long-lived systemic immune response to all PLVF vaccine antigens, which most likely is supported by memory cell response. On day 210, after primary immunization with a booster on day 30, the levels of IgA seroconversion remained high. There was a tendency for the persistance of the levels of four-fold seroconversion.

A long-lasting monovalent IgA response of up to 140 days was recorded after three immunizations with the semisynthetic *S. flexneri* 2a polysaccharide conjugate to the tetanus toxoid vaccine [26].

Polyvalent mucosal immune response to five LPS *Shigella* antigens in the human gastrointestinal tract was registered for the first time. One month after a single subcutaneous immunization with 125 μg of PLVF, LPS-specific IgA, IgG, and sIgA were detected in the saliva and feces of volunteers. Approximately 33% of volunteers had a two-fold increase in sIgA against *S. flexneri* 1b, 2a, 3a, and Y in stool and about 80% in saliva (see Table 4).

Our data can be interpreted in light of the hypothesis of J.B. Robbins, that after parenteral administration of a vaccine, antibodies specific to LPS can inactivate bacterial inoculum on pre-sensitized human epithelial surfaces [15]. In this regard, an immunogen like PLVF with a clear potency for inducing an OAg-specific IgA response on the gastrointestinal mucosa (especially in the oral cavity and colon) should be considered the most promising vaccine candidate for field protection against shigellosis.

## 5. Conclusions

Good tolerability of PLVF and pronounced humoral immune response allow us to proceed to phase 3 clinical trial stage.

## Figures and Tables

**Table 1 vaccines-13-00209-t001:** Clinical trial participant flow information.

Study Day	Events
14 days before the first dose	Obtaining written informed consent from the participant prior to inclusion in the study. Evaluation for inclusion criteria. Evaluation for exclusion criteria. Obtain medical history. Physical examination (includes examination of the skin, peripheral lymph nodes, auscultation of the lungs, heart, examination of abdominal organs, and assessment of neurological status). Measurement of vital signs (blood pressure, heart rate, and respiratory rate). Thermometry (in the armpit). Pregnancy test (women). Breath alcohol test. Urine drug test. Taking a blood sample for complete blood count and biochemical blood test, serological screening for HIV, syphilis, hepatitis B, and C, and determination of antibody levels to *S. flexneri*. The total volume of whole blood should be at least 10 mL, of which at least 5 mL should be used to determine antibody levels to *S. flexneri*. Urinalysis.
Month 0, Day 1 Hospitalization Randomization Vaccination 1	Physical examination. Measurement of vital signs. Pregnancy test (women). Breath alcohol test. Urine drug test. Evaluation for inclusion/exclusion criteria. Hospitalization for 24 h. Randomization. Thermometry pre-vaccination. Vaccination (subcutaneous injection of the vaccine into the upper arm). Thermometry post-vaccination (after 1–2 h, 7–8 h, and 12–13 h). Assessment of local reactions to vaccine administration (after 20 min and 5–8 h). Measurement of vital signs (after 1–2 h, 7–8 h, and 12 h). Obtaining fecal (1 g) and saliva (8 mL) samples.
Month 0, Day 2	Evaluation for exclusion criteria. Physical examination. Assessment of local reactions to vaccine administration. Thermometry. Measurement of vital signs. Taking a blood sample for complete blood count and biochemical blood test. Urinalysis. Handing out self-observation diaries. Instructing on filling out self-observation diary. Assessment of the presence of adverse events (AEs).
Month 0, Days 3–6	Filling out self-observation diaries.
Month 0, Day 7	Evaluation for exclusion criteria. Physical examination. Assessment of local reactions to vaccine administration. Thermometry. Measurement of vital signs. Taking a blood sample for complete blood count and biochemical blood test. Collection and verification of self-observation diaries. Assessment of the presence of AEs.
Month 0, Days 28–30	Filling out self-observation diaries.
Month 1, Day 1 Hospitalization Randomization Vaccination 2	Evaluation for inclusion criteria. Evaluation for exclusion criteria. Pregnancy test (women). Breath alcohol test. Urine drug test. Thermometry before the second dose of vaccine. Physical examination before the second dose of vaccine. Measurement of vital signs before the second dose of vaccine. Assessment of the presence of AEs. Taking a blood sample for complete blood count and biochemical blood test and determination of antibody levels to *S. flexneri*. The total volume of whole blood should be at least 10 mL, of which at least 5 mL should be used to determine antibody levels to *S. flexneri*. Obtaining fecal (1 g) and saliva (8 mL) samples Hospitalization for 24 h. Vaccination (subcutaneous injection of the vaccine into the upper arm).
Month 1, Day 2	Evaluation for exclusion criteria. Physical examination. Assessment of local reactions to vaccine administration. Thermometry. Measurement of vital signs. Taking a blood sample for complete blood count and biochemical blood test. Urinalysis. Handing out self-observation diaries. Assessment of the presence of AEs.
Month 1, Days 3–6	Filling out self-observation diaries.
Month 1, Day 7	Evaluation for exclusion criteria. Physical examination. Assessment of local reactions to vaccine administration. Thermometry. Measurement of vital signs. Taking a blood sample for complete blood count and biochemical blood test. Collection and verification of self-observation diaries. Assessment of the presence of AEs.
Month 1, Days 8–30	Filling out self-observation diaries.
Month 2, Day 1	Evaluation for exclusion criteria. Physical examination. Thermometry. Measurement of vital signs. Taking a blood sample for complete blood count and biochemical blood test and determination of antibody levels to *S. flexneri*. The total volume of whole blood should be at least 10 mL, of which at least 5 mL should be used to determine antibody levels to *S. flexneri*. Urinalysis. Assessment of the presence of AEs.
Month 7, Day 1	Evaluation for exclusion criteria. Physical examination. Thermometry. Measurement of vital signs. Taking a blood sample for complete blood count and biochemical blood test and determination of antibody levels to *S. flexneri*. The total volume of whole blood should be at least 10 mL, of which at least 5 mL should be used to determine antibody levels to *S. flexneri*. Urinalysis. Assessment of the presence of AEs. End of observation

**Table 2 vaccines-13-00209-t002:** Average demographic and anthropometric characteristics of all volunteers participating in the screening procedure and meeting the inclusion criteria for the study.

Group		Age	Height, cm	Weight, kg
All participants *n* = 80	Mean	44.5 ± 9.4	166.2 ± 6.2	72.2 ± 10.2
	Range	21.0–55.4	155–184	49–96
Group 1 (dose 62.5 µg) *n* = 40	Mean	45.7 ± 8.9	167.0 ± 6.1	74.1 ± 10.8
	Minimum	21.0	156	55
	Maximum	55.4	184	96
Group 2 (dose 125 µg) *n* = 40	Mean	43.3 ± 9.8	165.5 ± 6.3	70.4 ± 9.4
	Minimum	21.6	155	49
	Maximum	55.3	184	89

**Table 3 vaccines-13-00209-t003:** Proportion of participants (%) with seroconversion of *Shigella* LPS-specific IgA, IgG, and IgM.

Group	Ig Class	LPS Serotype	Seroconversion (%)		
			1 Mo After Vaccination 1	1 Mo After Vaccination 2	6 Mo After Vaccination 2
Group 1 (dose 62.5 µg)	IgA	*S. flexneri* 1b	42.5	30.0	35.0
		*S. flexneri* 2a	57.5	52.5	67.5
		*S. flexneri* 3a	35.0	40.0	52.5
		*S. flexneri* 6	32.5	32.5	50.0
		*S. flexneri* Y	45.0	37.5	42.5
	IgM	*S. flexneri* 1b	50.0	27.5	30.0
		*S. flexneri* 2a	47.5	30.0	30.0
		*S. flexneri* 3a	42.5	17.5	27.5
		*S. flexneri* 6	27.5	15.0	22.5
		*S. flexneri* Y	37.5	17.5	22.5
	IgG	*S. flexneri* 1b	35.0	35.0	40.0
		*S. flexneri* 2a	50.0	62.5	70.0
		*S. flexneri* 3a	42.5	42.5	52.5
		*S. flexneri* 6	25.0	40.0	60.0
		*S. flexneri* Y	47.5	40.0	60.0
Group 2 (dose 125 µg)	IgA	*S. flexneri* 1b	75.0	67.5	60.0
		*S. flexneri* 2a	82.5	80.0	72.5
		*S. flexneri* 3a	67.5	65.0	67.5
		*S. flexneri* 6	72.5	70.0	70.0
		*S. flexneri* Y	70.0	70.0	67.5
	IgM	*S. flexneri* 1b	55.0	47.5	42.5
		*S. flexneri* 2a	52.5	27.5	37.5
		*S. flexneri* 3a	52.5	35.0	35.0
		*S. flexneri* 6	27.5	20.0	25.0
		*S. flexneri* Y	40.0	32.5	27.5
	IgG	*S. flexneri* 1b	70.0	65.0	60.0
		*S. flexneri* 2a	77.5	72.5	70.0
		*S. flexneri* 3a	60.0	57.5	67.5
		*S. flexneri* 6	65.0	57.5	57.5
		*S. flexneri* Y	70.0	70.0	57.5

**Table 4 vaccines-13-00209-t004:** Proportion of participants (%) with ≥2- and ≥4-fold increases in *Shigella* LPS-specific IgA, IgG, and sIgA in stool and saliva one month after a single immunization with 125 µg of PLVF.

Group	Ig Class	LPS Serotype	Titer Increase in Stool (*n* = 15), %		Titer Increase in Saliva (*n* = 9), %	
			≥2	≥4	≥2	≥4
Group 2 (dose 125 µg)	IgA	*S. flexneri* 1b	26.7	26.7	44.4	11.1
		*S. flexneri* 2a	40.0	33.3	55.5	33.3
		*S. flexneri* 3a	40.0	33.3	33.3	22.2
		*S. flexneri* 6	40.0	26.7	33.3	33.3
		*S. flexneri* Y	53.3	46.7	77.8	33.3
	sIgA	*S. flexneri* 1b	26.7	26.7	66.7	22.2
		*S. flexneri* 2a	33.3	33.3	77.8	44.4
		*S. flexneri* 3a	40.0	33.3	77.8	44.4
		*S. flexneri* 6	33.3	13.3	88.9	44.4
		*S. flexneri* Y	40.0	26.7	88.9	55.5
	IgG	*S. flexneri* 1b	33.3	20	77.8	22.2
		*S. flexneri* 2a	40.0	33.3	77.8	55.5
		*S. flexneri* 3a	46.7	26.7	66.7	44.4
		*S. flexneri* 6	33.3	26.7	77.8	44.4
		*S. flexneri* Y	46.7	40	66.7	44.4

## Data Availability

The data presented in this study are available from the corresponding author upon reasonable request.

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
