# Peer review of "A Candidate Ac3-S-LPS Vaccine Against S. flexneri 1b, 2a, 3a, 6, and Y Activates Long-Lived Systemic and Mucosal Immune Responses in Healthy Volunteers: Results of an Open-Label, Randomized Phase 2 Clinical Trial"

_vaccines, 2025, doi:10.3390/vaccines13030209_

Round 1

Reviewer 1 Report

Comments and Suggestions for Authors

Authors conducted an open-label, phase-s clinical trial testing the safety and immunogenicity of a pentavalent candidate vaccine (AC3-S-LPS) against 5 serotypes of S. flexneri strains.  Groups were subcutaneously immunized with two dosages of antigens; IgA, IgA, IgM antibodies were determined from blood, fecal and saliva samples. 

Specific Comments:

1) How are antibodies titers defined?

2) How is seroconversion rate determined?

3) What is the purity of antigen used?  adjuvant?

Author Response

1) How are antibodies titers defined?

We thank the reviewer for careful review of our manuscript and appreciate his comments and suggestions

We added this information to the text (Lines 157-158). “The titer was determined using cut-off = M (blank, BSA adsorbed on plate) + 3SD.”

2) How is seroconversion rate determined?

We defined seroconversion as a ≥4-fold increase in specific antibody titers in serum compared to baseline.

We added this information to the text (Lines 155-157). “The immune response in volunteers is represented by a ≥4-fold increase in specific antibodies titers in serum (seroconversion) and a ≥2-fold and ≥4-fold increase in specific antibodies titers in saliva and coprofiltrates, compared to baseline.”

3) What is the purity of antigen used? adjuvant?

Vaccine was manufactured based on Ac3-S-LPS compounds from S. flexneri 1b, 2a, 3a, 6, and Y (PLVF) as the active substances at a dose of 125 μg (25 μg of each antigen compound) and contained the following formulation excipients: phenol (preservative) 0.75 mg, NaCl 4.15 mg, Na2HPO4 0.052 mg, and NaH2PO4 0.017 mg, and 0.5 mL sterile pyrogen-free water for injection. Vaccine preparation did not contain any adjuvant. Antigen composition and purity analysis was described in our previous paper (Ledov et al. 2023).

Reviewer 2 Report

Comments and Suggestions for Authors

This is an interesting study involving human subjects (volunteers) following up the previous report by the same group regarding the development of the vaccine and its administration in experimental animals. The study was well designed and conducted and the results are explicitely described. It is a bit surprising that no even minor side effects of the vaccine were recorded.

Author Response

It is a bit surprising that no even minor side effects of the vaccine were recorded.”

We would like to thank the reviewer for his/her positive comments on the manuscript. In addition, we would like to point out that we managed to achieve such low toxicity by controlled detoxification of LPS, which became apyrogenic but retained its immunogenic properties. This process was described in our previous publications about preclinical studies and phase 1 clinical trial of Shigella Ac3-S-LPS (Ledov et al. 2019, Ledov et al. 2023).

Reviewer 3 Report

Comments and Suggestions for Authors

The study conducted by Ledov et al. presents findings on the safety and immunogenicity of a subcutaneous pentavalent tri-acylated LPS Shigella flexneri vaccine in healthy volunteers. The trial design is appropriate; however, a more in-depth discussion of the results would strengthen the manuscript and enhance reader engagement. The absence of systemic or local adverse effects is a significant strength, reinforcing the tolerability of this vaccine. Nevertheless, several points of the manuscript require clarification and revision:

Specific comments

1. The authors state that IgG, sIgA and IgA levels are maintained long-term, but no significant booster effect is observed following the second dose (one month after the initial immunization). Could the authors offer a possible explanation for this lack of booster response? Might it be infuenced by factors such as the dosing interval, the vaccine formulation, or the limited sample size used to measure antibody levels in stool (n=15) and saliva (n=9)?

2. The results indicate no significant changes in blood and urine analyses. Despite, some deviations are observed. These should be clearly indicated in the corresponding supplementary tables.

3. Have the authors considered measuring the levels of inflammatory mediators, such as calprotectin or lactoferrin, in stool supernatants before and after immunization? If not, please clarify the reasoning behind this decision.

4. Given that immunogenicity was assessed using fecal and saliva samples, please specify whether these methods have been validated for this specific vaccine. In addition, provide a more detailed explanation of the methodology used to prepare the coprofiltrates, including whether commercial ELISA kits were used and which ones.

5. Please, explain the criteria used to define a positive seroresponse to vaccination among trial participants in each group.

6. Table 3. A better reorganization of the data would facilitate the comparison of seroconversion rates between Group 1 and 2.

7. Table 4 and lanes (202-209). Some data indicate a titer increase (%) in saliva greater than 2-fold and 4-fold for sIgA and IgG, both specific to S. flexneri 2a, 3a, 6 and Y, with combined values exceeding 100%. Please, could the authors provide an explanation for this observation?

6. The Discussion include some paragraphs that would be more appropriate in the Results section. Furthermore, a more comprehensive discussion, including comparisons with similar trials, would be beneficial.

7. Lanes (242-247): Do the seroconversion data refer to those presented in Table 3? If yes, please reference Table 3 in the text. Similarly, for lanes (261-266), do these refer to Table 4? Additionally, what do lanes (267-270) correspond to? Please clarify these references within the text.

Minor comments

1. Please, define the abbreviations AEs (adverse effects), GEMS (Global Enteric Multicenter Study) and MALT (mucosa-associated lymphatic tissue) upon their first use in the text.

2. The information in the “Registration” section is repeated in the “Institutional Review Board Statement” and should be revised accordingly.

Author Response

Reviewer #3:

  1. The authors state that IgG, sIgA and IgA levels are maintained long-term, but no significant booster effect is observed following the second dose (one month after the initial immunization). Could the authors offer a possible explanation for this lack of booster response? Might it be influenced by factors such as the dosing interval, the vaccine formulation, or the limited sample size used to measure antibody levels in stool (n=15) and saliva (n=9)?

We thank the reviewer for careful review of our manuscript and appreciate his insightful comments and suggestions.

The conclusion about the duration of the immune response is based on the results of a study of sera of immunized volunteers, where high titers of specific antibodies remain 6 months after repeated immunization. A booster O-antigen response in humans is not recorded for vaccines against shigellosis flexneri, regardless of the type of vaccine (conjugate, vesicular). The absence of a booster effect is probably due to the nature of O-specific polysaccharide chains of Shigella flexneri of different serotypes. These types of antigens are able to activate pre-existing memory cells during primary immunization, regardless of the type of vaccine antigen: conjugate, LPS, or OMV. Long-lived (during 6–7 months of observation) IgG and IgA human responses can be triggered by the involvement of memory cells. The administered dose of the vaccine (125 μg) is sufficient to induce an immune response in humans; this formulation was immunogenic in the first phase of trials. It is unlikely that changing the interval between vaccinations will change the dynamics of such a response. The vaccine induces the production of IgA, which, in our opinion, should contribute to the fight against infection already at the intestinal mucosa. We studied antibodies in the stool and saliva of vaccinated people only one month after a single immunization, with the goal of proving the appearance of specific antibodies on the gastrointestinal mucosa in principle.

Some aspects of this are briefly discussed on Lines 275-280.

  1. The results indicate no significant changes in blood and urine analyses. Despite, some deviations are observed. These should be clearly indicated in the corresponding supplementary tables.

We corrected normal hemoglobin levels for men and women upon contacting the clinical diagnostic laboratory. Thus, no deviations from the reference range or baseline levels were observed.

  1. Have the authors considered measuring the levels of inflammatory mediators, such as calprotectin or lactoferrin, in stool supernatants before and after immunization? If not, please clarify the reasoning behind this decision.

During phase 1 clinical study (results published in Russian Journal “Bacteriology”, abstract in English available at https://www.obolensk.org/bacteriology/new-number/item/522-ledov2024-9-4-p49-55)), we determined TNF-a, IL-1, IL-6, IL-12, INF-g, and cortisol blood levels in volunteers 2, 4, 6 hours after immunization with doses of 125 μg and 62.5 μg of vaccine and did not register significant changes. We would very much like to study the effect of such a unique antigen on the intestinal immune system, and not only that. However, the analysis of additional markers was associated with difficulties in obtaining permission to conduct the clinical trial from the Ministry of Health of the Russian Federation.

  1. Given that immunogenicity was assessed using fecal and saliva samples, please specify whether these methods have been validated for this specific vaccine. In addition, provide a more detailed explanation of the methodology used to prepare the coprofiltrates, including whether commercial ELISA kits were used and which ones.

The ELISA was performed according to a standard protocol, and validation of the method was provided to the Ministry of Health at the stage of obtaining permission for the clinical trial.

We added more information regarding ELISA and sample collection and processing to the Methods section (Lines 115-135).

  1. Please, explain the criteria used to define a positive seroresponse to vaccination among trial participants in each group.

The immune response in volunteers is represented by a ≥4-fold increase in specific antibodies (seroconversion) in serum compared to baseline. Lines 155-157

  1. Table 3. A better reorganization of the data would facilitate the comparison of seroconversion rates between Group 1 and 2.

We have a large data set because the vaccine is 5-valent, and we determined IgA, IgG, and IgM. Therefore, a table is the preferred way to present the data. Unfortunately, we did not find a better way of presenting data by reshuffling columns and rows.

We did not see a statistical difference between groups 1 and 2. We can only talk about a trend or a more pronounced response.

  1. Table 4 and lanes (202-209). Some data indicate a titer increase (%) in saliva greater than 2-fold and 4-fold for sIgA and IgG, both specific to S. flexneri 2a, 3a, 6 and Y, with combined values exceeding 100%. Please, could the authors provide an explanation for this observation?

For some serotypes, 2-fold antibody rises were detected in 70-80% vaccinated and 4-fold rises in 40-50% vaccinated. All volunteers with detected 4-fold antibody rises were simultaneously included in the vaccinated group with 2-fold antibody rises, and constitute therein approximately 50-60% of the total. Thus, summation of the ratios of 2- and 4-fold rises in antibodies is not applicable, as their sum may exceed 100%.

  1. The Discussion include some paragraphs that would be more appropriate in the Results section. Furthermore, a more comprehensive discussion, including comparisons with similar trials, would be beneficial.

We updated the Discussion section. We believe that these changes have improved the manuscript.

  1. Lanes (242-247): Do the seroconversion data refer to those presented in Table 3? If yes, please reference Table 3 in the text. Similarly, for lanes (261-266), do these refer to Table 4? Additionally, what do lanes (267-270) correspond to? Please clarify these references within the text.

Thank you for pointing this out. We modified the manuscript. Lanes 242-247 were deleted

For Lanes 261-266 text was added at new Lanes 287. Lanes 267-270 were deleted.

Minor comments

  1. Please, define the abbreviations AEs (adverse effects), GEMS (Global Enteric Multicenter Study) and MALT (mucosa-associated lymphatic tissue) upon their first use in the text.

The abbreviations were added to the text.

  1. The information in the “Registration” section is repeated in the “Institutional Review Board Statement” and should be revised accordingly.

Thank you for pointing this out. This issue was corrected.